# Quantile Autoencoder for Anomaly Detection

**Hogeon Seo[1*], Seunghyoung Ryu[1*], Jiyeon Yim[2], Junghoon Seo[3], Yonggyun Yu[1]**

[1] Korea Atomic Energy Research Institute, 111, Daedeok-daero 989 beon-gil, Daejeon, Korea
[2] Telementalhealth and Foresight-curing Artificial Intelligence, 193 Munji-ro, Yuseong-gu, Daejeon, Korea
[3] SI Analytics, Co., 70, Yuseong-daero 1689beon-gil, Yuseong-gu, Daejeon, Korea
hogeony@hogeony.com, ashryu@kaeri.re.kr, jyyim2043@gmail.com, jhseo@si-analytics.ai, ygyu@kaeri.re.kr

## Abstract

Anomaly detection (AD) is an essential task in a variety of industrial fields. AD based on deep neural networks has shown effective performance. Most methods for deep anomaly detection (DAD) use *the difference between the input and reconstructed data* or *the distance from the center of the cluster defined by normal cases* as a measure of abnormality. However, these metrics do not consider the diversity of normal cases. We propose a quantile autoencoder (QAE) as a novel DAD method to consider the data-oriented uncertainty. QAE obtains the anomaly score from both the reconstruction error and the channel-wise data uncertainty that is the range of the two quantiles of the reconstruction distribution. This anomaly scoring makes the score distributions of the normal and abnormal samples farther apart by narrowing the width of the distributions, which contributes to the improvement of AD performance. The performance of the proposed QAE was verified with various datasets, and the results show higher performance compared to the benchmark results.

## Introduction

Anomaly detection (AD), also known as novelty detection or outlier detection, is a task that distinguishes anomalous cases in a collection or stream of data (Grubbs 1969) and has been used in diverse applications such as fraud detection (Pawar, Kalavadekar, and Tambe 2014; Porwal and Mukund 2018; Adewumi and Akinyelu 2017), network security (Lee 2017; Aoudi, Iturbe, and Almgren 2018; Kwon et al. 2019), video surveillance (Ravanbakhsh et al. 2017; Xu et al. 2015; Kiran, Thomas, and Parakkal 2018), medical diagnostics (Schlegl et al. 2017; Baur et al. 2018; Litjens et al. 2017), and sensing (Kuzin and Borovicka 2016; Zhao et al. 2017; Beghi et al. 2014; Ball, Anderson, and Chan Sr 2017; Mohammadi et al. 2018). Recent advances in deep learning have had a substantial impact on the field of AD. Deep anomaly detection (DAD) has shown improved performance in many complicated AD tasks (Chalapathy and Chawla 2019).

The DAD can be based on supervised, unsupervised, and semi-supervised learning. When both normal and abnormal data are sufficient and labeled, the supervised AD is available (Gu, Akoglu, and Rinaldo 2019). Since the abnormal samples are insufficient and unlabeled in many applications (Chalapathy and Chawla 2019), the semi-supervised or unsupervised AD have been widely applied and the models learn lower-dimensional representative features relevant to normality by reconstructing normal data via autoencoders (Candès et al. 2011; Zou, Hastie, and Tibshirani 2006; Schölkopf, Smola, and Müller 1997; Pang et al. 2018; Pevnỳ 2016; Li, Hastie, and Church 2006). The anomaly score can then be obtained from the difference between the input and reconstructed data or the distance. The representative features can be learned. An anomaly can be detected by checking whether the anomaly score exceeds the threshold.

Adopting data uncertainty additionally for anomaly scoring via DAD is necessary because the normal data may have specific variations. There are two types of uncertainty: *epistemic uncertainty* and *aleatoric uncertainty* (Kendall and Gal 2017). Epistemic uncertainty, also known as reducible uncertainty or model uncertainty, comes from the deviation in model parameters in the deep learning process. Aleatoric uncertainty originates from the data itself and thus is inherent as well as irreducible. That is why it is also known as irreducible uncertainty or data uncertainty. Adopting the epistemic uncertainty has enhanced AD performance, such as several DAD models using the Monte-Carlo (MC) dropout (Zhu and Laptev 2017; Legrand, Trannois, and Cournier 2019; Leibig et al. 2017; Seeböck et al. 2019; Collin and De Vleeschouwer 2020). However, utilizing the aleatoric uncertainty in anomaly scoring has received less attention in the field of DAD though it was used as a threshold for the classification of normal and abnormal data in (Xu and Chen 2020).

In this study, we propose a novel DAD framework that introduces a quantile autoencoder (QAE) to consider an aleatoric uncertainty term. The uncertainty considered in the QAE is the range between two quantiles of the reconstruction distribution with the assumption of the channel-wise consistency in normal data, i.e., normal data deviate relatively less than abnormal data. The reconstructed data from the abnormal sample are prone to have a higher channel-wise uncertainty than that from the normal sample after the QAE is trained for normal samples. The effectiveness of QAE considering the reconstruction error and channel-wise data uncertainty was investigated with various datasets. The proposed QAE showed significant performance improve-

---

*These authors contributed equally.

ment in AD.

## Proposed Methodology

### Motivation

The motivation behind the proposed DAD framework, QAE, is as follows:

- **Use of uncertainty.** Due to the benefits of the ensemble, the more diverse the sources used in anomaly scoring, the better the performance in AD. Aleatoric uncertainty is one of the effective candidates able to be added in anomaly scoring for DAD.

- **Use of aleatoric uncertainty.** If the AE is capable of learning the features of normality in a latent space $z$, the reconstruction from $z$ will have a specific level of aleatoric uncertainty for each channel of the normal data. However, since the network is not exposed to the abnormal data, the channel-wise consistency under abnormal data may exceed that level for normal data. This distinction can contribute to the performance improvement in AD.

- **Use of a QAE.** The aleatoric uncertainty can be adopted by training a QAE model to output multiple quantiles simultaneously, which has the similar benefits to multi-task learning because of the nonlinear relationship between the quantiles.

### Quantile Autoencoder for Deep Anomaly Detection

Several approaches for dealing with uncertainty within a deep learning framework have been shown, including Bayesian deep learning, the MC dropout, and deep ensembles (Abdar et al. 2020). The MC dropout was used in uncertainty-based DAD (Zhu and Laptev 2017; Legrand, Trannois, and Cournier 2019; Leibig et al. 2017; Seeböck et al. 2019; Collin and De Vleeschouwer 2020). By applying dropout in both the training and inference stages. the neural network generated different outputs based on the probabilistic connections between the neurons. MC sampling was used to obtain the statistical information of the outputs. The dropout was activated during the inference stage. This approach mainly aims to exploit epistemic uncertainty.

The proposed QAE is designed to adapt an aleatoric uncertainty term. The QAE is based on the concept that the reconstructed data from the normal data will be within a specific range of variations for each channel of data, and the consistency of each channel is independently valid. The variation ranges of the abnormal data will be higher than those of the normal data because the reconstruction uses *the normal-fitted latent features*. The QAE uses not only the reconstruction error but also the range between two quantiles as a degree of uncertainty as an aleatoric uncertainty term by training the neural network. The anomaly score is finally derived from both the reconstruction error and the the aleatoric uncertainty term, as illustrated in Fig. 1.

**Quantile Autoencoder**   The proposed QAE is a variant of the AE that predicts the quantiles of the reconstruction distribution by minimizing the sum of pinball losses on multiple quantiles, whereas the basic AE predicts only the mean

of the reconstruction distribution by minimizing mean absolute error (MAE) or mean square error (MSE). Since the QAE performs multiple quantile regressions with a single AE model, it can be regarded as multi-task learning.

Let $X$ be a random variable with a cumulative distribution function $F_X(x)$. The $\tau$-quantile $x_\tau$ is $F_X^{-1}(\tau)$, where $\tau \in (0, 1)$. The proposed QAE $Q$ consists of an encoder $Q_{enc}$ and a quantile decoder $Q_{dec}$ as follows:

$$Q_{enc}(x) = z, \tag{1}$$

$$Q_{dec}(z) = [x_l, x_m, x_u] = \mathbf{x}_\tau, \tag{2}$$

where $l < 0.5$, $u > 0.5$, and $m = 0.5$ are the lower, upper, and median quantiles, respectively.

Quantile regression is conducted by minimizing the pinball loss (Steinwart, Christmann et al. 2011). For the $\tau$-quantile ($x \geq x_\tau$), the pinball loss is defined as follows:

$$L_\tau(x_\tau, x) = \tau(x - x_\tau), \tag{3}$$

where $x$ and $x_\tau$ are an input and a prediction of $\tau$-quantile, respectively. For the $\tau$-quantile ($x < x_\tau$), the pinball loss is defined as follows:

$$L_\tau(x_\tau, x) = (1 - \tau)(x_\tau - x). \tag{4}$$

Therefore, the pinball loss function is a tilted absolute error function weighted by the target $\tau$. Intuitively, the pinball loss results in a higher penalty of overestimation ($x_\tau \geq x$) for low quantiles ($\tau < 0.5$). Therefore, the network is trained to underpredict. Similarly, the network overpredicts for high quantiles $\tau > 0.5$. Finally, the loss is calculated using the multiple quantile loss for the proposed QAE $L_Q$, which is a summation of the pinball losses with multiple $\tau$s:

$$L_Q(\mathbf{x}_\tau, x) = L_l(x_l, x) + L_m(x_m, x) + L_u(x_u, x). \tag{5}$$

**Anomaly Scoring in QAE**   The anomaly score in QAE is derived from both the reconstruction error and aleatoric uncertainty term measured by the range between the predicted upper and lower quantiles. From the output of the QAE $\mathbf{x}_\tau$, the reconstruction error $\epsilon_{rec}$ and uncertainty $\epsilon_{unc}$ in the form of row vector are defined as follows:

$$\epsilon_{rec} = x - x_m, \tag{6}$$

$$\epsilon_{unc} = x_u - x_l. \tag{7}$$

The anomaly score $A_R(x)$ and quantile-based anomaly score $A_Q(x)$ with $\epsilon = [\epsilon_{rec}, \epsilon_{unc}]$ are expressed as follows:

$$A_R(x) = (\sqrt{\epsilon_{rec}\epsilon_{rec}^T})^p / d_{\epsilon_{rec}}, \tag{8}$$

$$A_Q(x) = (\sqrt{\epsilon\epsilon^T})^p / d_\epsilon, \tag{9}$$

where $d$ is the dimension of the corresponding vector, and $p$ represents either 1 or 2 for MAE or MSE, respectively. Because $\epsilon_{rec}$ and $\epsilon_{unc}$ exist in different domains, the difference between their magnitude ranges has a negative impact on anomaly scoring. A similar problem was reported (Kim et al. 2019) when utilizing reconstruction errors in both the original space and the latent spaces. To match the range of the error terms, a normalized distance with two-step orthogonalization and scaling was introduced, and it contributed

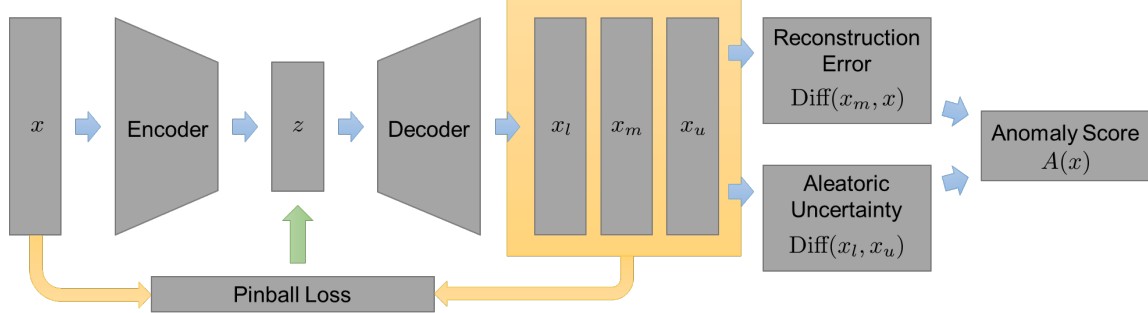

Figure 1: The proposed QAE framework for AD. The QAE predicts the median value and the two quantiles for anomaly scoring with the reconstruction error and aleatoric uncertainty term.

| Modal | Dataset | $N$ | $d_x$ | $d_z$ | Class Count | Domain | Anomaly Target |
|-------|---------|-----|-------|-------|-------------|--------|----------------|
| Uni | RARM | 20,221 | 6 | 3 | 2 | Robotics | Malfunctions |
| Uni | NASA | 4,687 | 33 | 10 | 2 | Astronomy | Hazardous asteroids |
| Uni | MI-F | 25,286 | 58 | 23 | 2 | CNC Milling | Machine not completed |
| Uni | MI-V | 23,125 | 58 | 23 | 2 | CNC Milling | Workpiece out of specification |
| Uni | EOPT | 90,515 | 20 | 6 | 2 | Storage System | System Failures |
| Multi | SNSR | 58,509 | 48 | 17 | 11 | Electric Currents | Defective conditions |
| Multi | OTTO | 61,878 | 93 | 66 | 9 | E-commerce | Types of products |

*$N$: The Number of Data, $d_x$: The Dimension of Input Data, $d_z$: The Dimension of Bottleneck Features

Table 1: The information of the benchmark datasets.

| Dataset | AE $A_R$ $p=2$ | QAE $A_Q$ $p=1$ | $A_Q$ $p=2$ | $A_{QN}$ $p=1$ | $A_{QN}$ $p=2$ |
|---------|------|------|------|------|------|
| RARM | 68.7 | 78.2 | 78.1 | 80.3 | **80.4** |
| NASA | 71.9 | 68.8 | 71.0 | 72.6 | **72.7** |
| MI-F | 60.7 | 50.4 | 47.4 | 73.0 | **74.6** |
| MI-V | 89.7 | 68.4 | 69.7 | **92.3** | 92.0 |
| EOPT | 61.0 | **71.6** | 70.6 | 61.5 | 61.6 |
| SNSR | 61.3 | 59.9 | 62.4 | 66.6 | **67.3** |
| OTTO | 61.7 | **61.9** | 58.6 | 59.4 | 60.7 |

***Bold**: Best, Underline: Second Best

Table 2: Comparison of AUROC [%] of RAPP and QAE

to the alleviation of the difference problem as well as the performance improvement in AD. For the same purpose, the proposed QAE also calculates the normalized anomaly score $A_{QN}(x)$ based on the Mahalanobis distance as follows:

$$A_{QN}(x) = \left( \sqrt{(\epsilon - \mu)S^{-1}(\epsilon - \mu)^T} \right)^p, \quad (10)$$

where $\mu$ and $S$ are the channel-wise mean and covariance matrix obtained from $\epsilon$ of the training set, respectively.

## Experiments

To verify the effectiveness of the proposed QAE, the area under the receiver operating characteristic (AUROC) was in-

vestigated with various datasets. The verification framework and datasets are referred from RAPP (Kim et al. 2019).

## Datasets and Problem Settings

The datasets used to verify the performance of the proposed QAE is listed in Table 1 modified from RAPP (Kim et al. 2019). For unimodal datasets, the target class is the normal class. The QAE is trained with only the samples of the target class. For multimodal datasets, the performance was evaluated for each class as the target class and then their averaged performance was finally calculated. The target class is the abnormal class and only the samples of the remaining classes are used for training. In both the modal cases, 60% of the samples of the normal class is randomly selected and used for training. Each half of the remaining samples is used as the validation and test datasets, respectively. The test dataset additionally contains the samples of the abnormal class and used to calculate the AUROC in AD. It should be noted that all input features are normalized using the z-score normalization. Furthermore, several multi-dimensional point datasets from outlier detection datasets (ODDS) (Rayana 2016) were used to compare the AD performance with various methodologies from statistical to deep learning approaches.

## Network Structure and Experimental Setup

We built QAE using PyTorch (Paszke et al. 2019) based on the same backbone network structure of RAPP (Kim et al. 2019), except for the final layer and loss function. The QAE

| Method | Model | Dataset | | | | | | |
|--------|-------|---------|---------|-----------|--------|--------------|------------|--------|
| | | optdigits | pendigits | satellite | letter | featuredmnist | ionosphere | speech |
| Machine Learning | IForest | 71.5 | 96.2 | 68.6 | 60.0 | 79.3 | 84.2 | 44.2 |
| | LODA | 71.4 | 95.1 | 72.6 | 62.2 | 59.6 | 85.3 | 44.1 |
| | LOF | 61.2 | 85.1 | 57.9 | 84.2 | 84.0 | 90.0 | 47.9 |
| | DTM2 | 56.1 | 95.8 | 76.8 | 85.6 | 86.2 | 92.8 | 48.3 |
| | $k$NN | 53.7 | 95.0 | 76.5 | 86.2 | 86.1 | 92.8 | 48.3 |
| | $k_{th}$NN | 84.2 | 97.1 | 79.6 | 81.0 | 86.2 | 92.0 | 47.9 |
| | OCSVM | 55.8 | 93.5 | 65.0 | 55.7 | 83.5 | 81.2 | - |
| Deep Learning | DSVDD | 50.6 | 61.3 | 63.1 | 46.5 | 53.8 | 73.5 | - |
| | DAGMM | 29.0 | 87.2 | 66.7 | 43.3 | 65.2 | 46.7 | - |
| | SO-GAAL | 48.7 | 25.7 | 64.0 | 60.1 | 79.5 | 78.3 | - |
| | AE | 90.7 | 68.5 | 57.5 | 82.9 | 80.2 | 82.1 | - |
| | VAE | 76.8 | 93.1 | 60.3 | 51.7 | 84.7 | 76.0 | - |
| | RCA | 91.4 | 90.3 | 69.0 | 79.5 | 82.7 | 79.6 | - |
| | **QAE** | **96.9** | **99.1** | **86.4** | **89** | **90.6** | **95.8** | **55.5** |

\***Bold**: Best, Underline: Second Best

Table 3: Comparison of AUROC [%] of machine learning models, deep learning models, and QAE for various datasets.

was modified to provide the prediction of multiple quantiles $\mathbf{x}_\tau$. The target quantiles were set to $l = 0.3, m = 0.5$ and $u = 0.7$ in this study. Each $Q_{enc}$ and $Q_{dec}$ had ten layers with the Leaky-ReLU activation function. The dimensions of bottleneck features $d_z$ were derived by a principal component analysis (PCA), as listed in Table 1. The QAE was trained via Adam optimizer (Kingma and Ba 2014). Note that these experimental setups are identical to those described in the work of (Kim et al. 2019).

## Results

The AD performance of the QAE ($A_Q, A_{QN}$) was compared to that of the AE ($A_R$) of RAPP (Kim et al. 2019) according to the anomaly scoring method: MAE ($p = 1$) and MSE ($p = 2$), as listed in Table 2. The proposed QAE ($A_Q, A_{QN}$) shows higher AUROC than the AE ($A_R$), and $A_{QN}$ with MSE ($p = 2$) especially shows the best performance in most of the cases. These support that using both the reconstruction error and the aleatoric uncertainty term in anomaly scoring improves the AD performance. Moreover, the normalization in anomaly scoring contributes to better AD performance improvement. In particular, significant improvements was observed for the sensor-related datasets: MI-F, MI-V, RARM, and SNSR. They were collected by real-valued sensor measurements (e.g., position, velocity, voltage, and current). This supports that the proposed QAE can contribute to performance improvement of DAD in many real-world industrial applications.

In addition, the proposed QAE with $A_{QN}(p = 2)$ achieved the best performance compared to various anomaly detection methodologies from statistical to recent deep learning approaches, as listed in Table 3. The proposed QAE shows robust DAD performance compared to other neural network-based models. Note that the results of six methods (IForest, LODA, LOF, DTM2, $k$NN, and $k_{th}$NN) were borrowed from (Gu, Akoglu, and Rinaldo 2019) and seven mod-

els (OCSVM, DSVDD, DAGMM, SO-GAAL, AE, VAE, and RCA) were borrowed from (Liu et al. 2021), respectively.

## Conclusions

This research was motivated by diversifying the sources for anomaly scoring to improve the DAD performance with a single neural network. To take advantage of aleatoric uncertainty for AD, we propose a novel DAD framework, QAE, that predicts not only median but also quantiles of reconstruction distribution. The reconstructed output from the abnormal data is likely to have larger channel-wise uncertainty than that from the normal data after training the QAE with only the normal samples. Therefore, adopting a channel-wise uncertainty term additionally in anomaly scoring can contribute to improve the AD performance. The effectiveness of the proposed QAE is verified with various datasets. The proposed QAE ($A_Q, A_{QN}$) shows higher AUROC than the AE ($A_R$), which support that using both the reconstruction error and the aleatoric uncertainty term in anomaly scoring improves the AD performance. Furthermore, the $A_{QN}$ with ($p = 2$) especially shows the best performance in most of the cases, which means that the normalization in anomaly scoring contributes to better AD performance improvement. The effectiveness of the QAE is also supported by superior performance compared to diverse anomaly detection methodologies, from statistical to recent deep learning approaches. Since AD performance improvement was clearly observed for the sensor-related datasets, the proposed QAE is expected to be advantageous for DAD in a wide variety of real-world industrial applications.

## Acknowledgments

This research was supported by a grant from Korea Atomic Energy Research Institute (KAERI) R&D Program (No.

KAERI-524450-21) and the National Research Foundation of Korea (NRF) grant funded by the Korea government(MSIT) (NRF-2021R1F1A1051290).

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
