# OpenReview forum: "Quantile Autoencoder for Anomaly Detection"
_AAAI.org/2022/Workshop/ADAM — AAAI 2022 Workshop ADAM_

### Official Review · Reviewer_7P6j · 2021-11-30
**Good work; need more comparisons**

**Rating:** 6
**Confidence:** 3

**Review:**

In this paper, the authors propose an autoencoder-based anomaly detection framework specific to aleatoric uncertainty terms using a quantile autoencoder. Apart from the reconstruction error being a metric for anomaly detection, the authors also use the channel-wise consistency to further get two quantiles of the reconstruction distribution for measuring the distribution. The idea is interesting and the results are shown to be significantly better than the benchmark considered (i.e. the vanilla autoencoders).

The only concern in my opinion on this paper is that the comparison is mainly performed with AE (which is sufficient for a workshop but maybe not for eventual publication). I suggest the authors look for comparing the method with other baselines including (but not limited to):

1. https://arxiv.org/pdf/2103.12051.pdf
2. https://www.dbs.ifi.lmu.de/Publikationen/Papers/LOF.pdf
3. https://openreview.net/forum?id=rctFXFsFvbI

---

### Official Review · Reviewer_yxod · 2021-12-01
**Very interesting approach for adapting autoencoders to anomaly detection**

**Rating:** 8
**Confidence:** 4

**Review:**

The paper proposes the quantile autoencoder (QAE) as an architecture for deep anomaly detection. The main idea is to use a "pinball loss" that also measures the quality of quantiles of the output distribution (as opposed to standard autoencoders that only predicts the expected value). Results are shown on anomaly detection using various benchmark datasets.

The paper is very well written and the contributions are timely.